# Nutrients, Diet, and Other Factors in Prenatal Life and Bone Health in Young Adults: A Systematic Review of Longitudinal Studies

**DOI:** 10.3390/nu12092866

**Published:** 2020-09-19

**Authors:** Karina H. Jensen, Kamilla R. Riis, Bo Abrahamsen, Mina N. Händel

**Affiliations:** 1Department of Medicine, Slagelse Hospital, 4200 Slagelse, Denmark; Karina.Haar.Jensen@gmail.com; 2Department of Endocrinology and Metabolism, Odense University Hospital, 5000 Odense C, Denmark; Kamilla.Ryom.Riis@gmail.com; 3Department of Medicine, Holbæk Hospital, 4300 Holbæk, Denmark; B.abrahamsen@physician.dk; 4Institute of Clinical Research, OPEN-Odense Patient Data Explorative Network, University of Southern Denmark, 5000 Odense, Denmark; 5The Parker Institute, Bispebjerg and Frederiksberg Hospital, 2000 Frederiksberg, Denmark

**Keywords:** bone health, bone mineral density, prenatal exposure, prenatal nutrition, peak bone mass, systematic review

## Abstract

Optimizing skeletal health in early life has potential effects on bone health later in childhood and in adulthood. We aimed to evaluate the existing evidence that maternal exposures during pregnancy have an impact on the subsequent bone health among offspring in young adults aged between 16 and 30 years. The protocol is registered in the International Prospective Register of Systematic Reviews (PROSPERO) (ID: CRD42019126890). The search was conducted up to 2 April 2019. We included seven observational prospective cohort studies that examined the association between maternal dietary factors, vitamin D concentration, age, preeclampsia, and smoking with any bone indices among offspring. The results indicated that high concentrations of maternal vitamin D; low fat intake; and high intakes of calcium, phosphorus, and magnesium may increase the bone mineral density in offspring at age 16. Evidence also suggests that the offspring of younger mothers may have a higher peak bone mass. It remains inconclusive whether there is an influence of preeclampsia or maternal smoking on bone health among young adults. Our assessment of internal validity warrants a cautious interpretation of these results, as all of the included studies were judged to have serious risks of bias. High-quality studies assessing whether prenatal prognostic factors are associated with bone health in young adults are needed.

## 1. Introduction

Osteoporosis causes more than 8.9 million fractures annually worldwide, of which more than 4.5 million occur in America and Europe [1]. The overall lifetime risk for a wrist, hip, or vertebral fracture has been estimated to be 30–40% in developed countries [1]. Thus, osteoporosis gives rise to a high global societal and individual disease burden and is associated with significant morbidity and mortality and a reduction in the quality of life [2].

Scientific contributions to the life course perspective of the prevention of osteoporosis have, over the last few decades, been growing considerably. In the Baker hypothesis, it was postulated that the prenatal environment induces long-lasting changes in physiological parameters such as hormonal levels, glucose tolerance, and satiety, a phenomenon known as “development programming”. The hypothesis indicates that this programming allows the offspring to use “cues” from the mother to adapt to the most likely postnatal environment [3].

This hypothesis is supported by studies evaluating the impact of prenatal exposure on bone health in children and teenagers. Maternal diet and 25(OH)-vitamin D status during pregnancy might be associated with offspring bone development at birth and in childhood. However, the findings are inconsistent [4,5,6,7,8,9,10,11,12]. Maternal cigarette smoking might also have the ability to modulate bone mineral acquisition during intrauterine life [13]. In a large study from Finland, elevated fracture risk was demonstrated in offspring at age seven from mothers who smoked during pregnancy [14]. However, there are conflicting results regarding bone health in children who have been exposed to maternal smoking during pregnancy [11,13]. An association between maternal age and the incidence of fractures in children at 11 years of age has also been demonstrated [15]. 

All of the above suggests that prenatal exposure may have an effect on bone health and fracture risk in children and teenagers—maybe with an influence on developing osteoporosis later in life. Since a low bone mass later in adult life is attributable to a lower than optimal peak bone accumulation in young adulthood and an accelerated loss of bone mass later in life [16,17], it may also be relevant to examine whether the associations between prenatal exposures and bone health persist into early adulthood. Thus, this systematic review aimed to summarize the currently available literature regarding the association between prenatal environmental exposures and bone health, in particular bone mineral density (BMD), in early adulthood in offspring, including to evaluate the research quality parameters.

## 2. Materials and Methods 

Before initiating the review, a protocol was developed following a pre-specified checklist and was reported in line with the PRISMA-P statement [18], and the guidelines from the Cochrane Collaboration, for details refer to Appendix A The study protocol was registered in the International Prospective Register of Systematic Reviews (PROSPERO). The protocol was submitted on 1 March 2019 and published on 3 May 2019 (ID: CRD42019126890) [19].

### 2.1. Search

An electronic comprehensive literature search was performed to identify the association between various fetal environmental exposures and the development of bone health in offspring during adulthood. The search was conducted on 2 April 2019 by Karina H. Jensen (KHJ) in MEDLINE (via PubMed), CENTRAL (via Cochrane Library), and EMBASE (via OVID) without time restrictions. The search strategy was developed using Medical Subject Headings (MeSH terms) and text words related to our eligibility criteria. The full search strategy, including MeSH terms and text words related to fetal life, exposure, and outcome, is compiled in Appendix A.

At the protocol stage, a preliminary search was conducted without any hedges, which resulted in a large degree of “noise” of irrelevant search results. As a consequence, we included some predefined search limits; studies dealing with “humans” and publications in English or Danish were of interest. In MEDLINE and EMBASE, the limits were extended to “Multicenter study”, “Randomized controlled trials” (RCT), “Clinical trial”, “Systematic review”, and “Meta-analysis”, as they were the studies of interest. The last added limit was the field “Title/Abstract”, related to the text availability. 

Reference lists of included articles and previous reviews were hand-searched, and content experts, authors Mina N. Händel(MNH) and Bo Abrahamsen(BA), were conferred with if any relevant references were missing. Any related references were included in the full-text screening.

### 2.2. Study Selection

All the studies were assessed and organized as references in the open-access software program Mendeley Desktop (version 1.19.4, Mendeley Ltd, London, United Kingdom). All the results from the search strategy were cross-checked for duplicates, and any duplicate was removed. After duplicates were removed, references were imported for screening in the Covidence systematic review software (Veritas Health Innovation, Melbourne, Australia; available at www.covidence.org). The Covidence software was initially used for the title and abstract screening primarily by two reviewers independently of one another (KHJ and Kamilla R. Riis (KRR), with help from a third reviewer MNH in the process) according to the eligibility criteria. Later, the software was used for full-text screening. Potentially eligible studies were retrieved, and the full texts were evaluated for eligibility by the same two reviewers (KHJ and KRR, supervised by MNH). Any disagreement was discussed and solved by consensus. If consensus could not be reached, the third reviewer (MNH) took part in the final decision process and had the final vote. Controversy about the extracted data between the two reviewers was discussed with a third reviewer (MNH), who had the deciding vote. 

Studies were included if outcomes were reported in offspring aged 16 to 30 years old and were born to mothers who were >18 years old during pregnancy. The measured outcomes of interest were any reported bone indices (BMD, bone mineral content (BMC), bone area (BA), peak bone mass (PBM), and trabecular bone score) or reported fractures (expressed as risk metrics, events, or event rates). There was no restriction on environmental exposure during pregnancy, thus including maternal lifestyle factors, anthropometry, medication, comorbidity, or pregnancy-related complications. If there were any additional relevant outcomes reported in the related studies, these were considered for incorporation. 

Randomized clinical trials and prospective or retrospective observational studies (cohort and case-control) were included. Cross-sectional studies, case series, case reports, animal studies, and non-original studies, including editorials, conference abstracts, and letters to the editor, were excluded. 

### 2.3. Data Collection

Relevant information from the included studies was extracted using a predefined data extraction template developed for this systematic review. From each identified study, we obtained data, including participant characteristics (e.g., age, gender, ethnicity), study characteristics (sample size, country, study design), outcome measures, and results. 

Where relevant, any missing or new data were sought from the authors of the studies obtained for this systematic review. The reply deadline from the authors was kept open throughout the conduction of the present review. We requested additional data from one author, but the author did not respond [20].

### 2.4. Quality Assessment

Since only non-randomized studies were included, the methodological quality of each study was assessed using the “Risk Of Bias In Non-randomized Studies of Interventions” tool (ROBINS-I) [21]. To evaluate the first domain (bias due to confounding), the evidence-based key confounders that we selected a priori comprised age, gender, ethnicity, and maternal body mass index (BMI) through a directed acyclic graph diagram. Co-interventions such as dietary counseling and contact with health nurses were predefined as possible confounders. A conclusion within each domain was reached by categorizing the risk of bias to either low, moderate, serious, or critical. This was done individually by one reviewer (KHJ or KRR). Afterward, the judgment was cross-checked, discussed, and reviewed with the other reviewer. If there were any discrepancies, a third reviewer (MNH) contributed and made the final decision. The quality of the evidence was assessed using the “Grades of Recommendation, Assessment, Development, and Evaluation” (GRADE) approach [22], which is categorized as very low, low, moderate, and high. According to GRADE, the certainty in the evidence starts at low in observational studies and is then assessed for possible downgrading based on the following domains: overall risk of bias, inconsistency, indirectness, imprecision, and publication bias. Only at substantial dose–response associations or when no risk of confounding is present can the possibility of upgrading be assessed. 

### 2.5. Meta-Analysis

Due to the scarcity of high-quality studies, a meta-analysis was deemed inapplicable, although this was planned in the protocol. Therefore, vote counting was used as a synthesis method, where we only assessed whether there was any evidence of an association rather than to evaluate the average effect size [23]. 

## 3. Results

### 3.1. Literature Search

Through the literature search up to 2 April 2019, we identified 3656 records after removing duplicates. However, the screening of titles and abstracts lead to the exclusion of 3440 records. A total of 233 articles were eligible for the full-text assessment. The full-text evaluation led to the exclusion of 230 records (see Appendix A for a list of excluded articles and reasons for exclusion). After internal discussion in the author group (content experts), we identified an additional four studies for inclusion that met the review criteria but had not been identified in the primary literature search [24,25,26,27]. In total, we identified seven eligible prospective distinct cohort studies [24,25,26,27,28,29,30]. A PRISMA flow diagram presenting data on the study selection process is illustrated in Figure 1.

### 3.2. Cohort Study Characteristics 

All seven studies were prospective cohort studies. The studies originate in different countries—i.e., Sweden, Finland, England, Australia, and Brazil. These studies are all relatively new and were published from 2010 to 2015. The sample size varied from 216 to 3088 participants. In one study, only men were included [30], whilst in the remaining six studies both genders were included. The mean age in the offspring was from 16.2 to 22.7 years old. Only two studies had information about ethnicity [27,29]; likewise, BMI was not obtained in all of the studies [24,26,28,30,31]. None of the seven included studies had conflicts of interest to declare. The detailed characteristics of the included studies are compiled in Table 1 and Appendix A. 

### 3.3. Risk of Bias within the Studies

The risk of bias for each study was assessed, and the results are presented in Table 2. The most problematic domain was bias due to missing data, where all of the included studies were rated with serious risk of bias. Within the domain of bias due to confounding, five of the studies were classified as having a serious risk of bias [25,26,27,28,30]. 

Except for the study by Miettola et al. [25], the studies did not indicate bias in the selection of reported results. Still, since none of the studies had reported a pre-specified protocol, we rated the risk of bias to be moderate within this domain. The studies by Yin et al. and Jones et al. [26,28] had serious risk of bias due to the selection of participants in the studies, while the rest of the studies had a low risk of bias due to this domain. The overall judgment of the risk of bias in all studies was rated as a serious risk of bias.

In summary, the observational studies included were graded as very low-quality evidence, due to their serious risk of bias. 

### 3.4. Qualitative Syntheses of Results

The exposures varied among the studies and included diet [28], vitamin D concentration in serum [29], preeclampsia [24,25], maternal smoking [26,27], and age [30]. As a common denominator, all seven studies evaluated BMD in the offspring [25,26,27,28,29,30]. Some studies also evaluated bone health with BA [29,30] and BMC [25,27,28,29,30], and one study also evaluated bone health with cortical cross-sectional area (CSA), areal bone mineral density (aBMD), volumetric bone mineral density (vBMD), and endosteal/periosteal circumference [30]. Besides this, one study measured bone mineral apparent density (BMAD) [25] and one study evaluated fractures in the offspring [26]. This review focused on BMD as the primary outcome in the syntheses of the results. 

### 3.5. Maternal Diet and Vitamin D 

Two articles addressing the possible effect of intrauterine nutrition on bone health in young adults met the criteria for inclusion. The results are compiled in Table 3. Yin et al. had complete data on 216 mother—offspring pairs from an initial cohort of 1435 Tasmanian mothers, intending to investigate sudden infant death syndrome [26]. The offspring were asked to participate in a study on bone mass at age 16 years, where a Dual-Energy X-ray Absorptiometry (DXA) scan was performed (measuring total body, femoral neck, and lumbar spine BMD). Zhu et al. included 341 mother–offspring pairs from the Western Australian Pregnancy Cohort (Raine RAINE) study; this cohort initially recruited 2900 pregnant women. A DXA scan was performed at the 20 years follow-up visit, and the outcome was total body bone measurements. 

#### 3.5.1. Total Body BMD

In the Tasmanian [28] study with 216 participants, no food groups or nutrients were found to be associated with total body BMD in either univariate or multivariate analyses (Table 3).

Zhu et al. [29] found that the maternal serum 25-hydroxyvitamin D (25OHD) was associated with the total body BMD in the offspring, in both a dose–response analysis as well as in the dichotomized analysis. In the dose–response analysis, each additional 10 nmol/L of maternal serum 25OHD was associated with an increase in BMD of 4.6 mg/cm^2^ (95% CI: 0.1–9.1). The analysis was adjusted for the following offspring factors—age at scan, gender, fat mass, lean mass and height—and for maternal factors: i.e., smoking, socioeconomic status, age, parity, and BMI (Table 3). 

In the dichotomized analysis, maternal vitamin D deficiency (serum 25OHD < 50 nmol/L) was associated with a 2.4% lower total body BMD (estimated mean ± s.e.: 1048 ± 8 versus 1074 ± 7 mg/cm^2^, *p* = 0.019) when adjusted for offspring age at scan and gender, a 2.4% lower total body BMD (1049 ± 9 versus 1074 ± 7 mg/cm^2^, *p* = 0.03) when further adjusted for maternal factors (age at scan, gender, maternal smoking, socioeconomic status, maternal age, and parity) and a 1.7% lower total body BMD (1053 ± 7 versus 1071 ± 5 mg/cm^2^, *p* = 0.04) in the fully adjusted model (offspring (fat mass, lean mass, height, age at scan, and gender) plus maternal factors (smoking, socioeconomic status, age, parity, and BMI)).

#### 3.5.2. Femoral Neck BMD

Yin and collaborators [28] found an inverse association between maternal fat density intake and femoral neck BMD (Table 3) after adjustment for offspring factors (gender, weight at age 16 years, sunlight exposure in the wintertime at age 16 years, sports participation, current calcium intake, Tanner stage, ever breastfed) and maternal factors (smoking during pregnancy and age at the time of childbirth). Femoral neck BMD was directly associated with magnesium density (*p* < 0.05) before adjustment (data not presented in the original article), but after adjustment for maternal and offspring factors, as previously defined in this Section 3.5.2, the association no longer remained. 

Further, femoral neck BMD was associated with the tertiles of fat and magnesium density (*p* < 0.05), showing that adolescents from mothers with lower intake of fat density and the highest intake of magnesium density had a 2.2% higher BMD in the femoral neck. 

In multivariate regression analysis, fat and magnesium density remained inversely and directly associated with the femoral BMD, respectively. In the study, the authors only included nutrients in the model if a *p*-value below 0.05 was detected. Which covariates were added to the model were not clearly described in the article. 

There were no associations between the maternal intake of protein, carbohydrate, calcium, or phosphorus density and the BMD in the femoral neck of the offspring. Maternal meat, fish, milk, vegetable, or fruit intake was not associated with the BMD in the femoral neck in an unadjusted model.

#### 3.5.3. Lumbar Spine BMD

In the Tasmanian cohort study [28], the investigators found an inverse association between the maternal fat density intake and the femoral neck BMD in offspring at age 16 years after adjustment for the offspring and maternal factors mentioned above (Table 3). On the contrary, a positive association between the maternal intake of magnesium and calcium density with BMD at the lumbar spine was found; data were adjusted for potential confounders, such as sex, weight at age 16 years, sunlight exposure in winter at age 16 years, sports participation, current calcium intake, Tanner stage, ever breastfed, smoking during pregnancy, and maternal age at the time of childbirth.

In a tertile-based analysis, the lumbar spine BMD was associated with the tertiles of phosphorus and magnesium density (*p* < 0.05), showing that adolescents with a lower intake of fat density and a higher intake density of calcium, phosphorus, and magnesium had a BMD 3.8% higher in the lumbar spine. 

Fat density remained inversely associated with the lumbar spine BMD in multivariate regression analysis, but calcium, magnesium, and phosphorus density were no longer associated with the lumbar spine BMD. Only nutrients with a *p*-value below 0.05 were examined. Again, the article does not state which additional covariates were adjusted for, besides the maternal and offspring factors (offspring sex, weight at age 16 years, sunlight exposure in winter at age 16 years, sports participation, current calcium intake, Tanner stage, ever breastfed, smoking during pregnancy, and maternal age at the time of childbirth). 

After adjustment for maternal and offspring factors and some additional confounders not explicitly stated in the article, maternal milk intake was directly associated with lumbar spine BMD, but the maternal intake of meat, vegetables, fish, or fruit was not associated with lumbar spine BMD [28].

### 3.6. Preeclampsia and Gestational Hypertension 

Two of the articles included in this systematic review evaluated the effects of preeclampsia (PE) and gestational hypertension (GH) on bone health in the offspring, results are summarized in Table 4. The study from Hannam et al. was a sub-cohort with 3088 mother–offspring pairs from the Avon Longitudinal Study of Parents and Children (ALSPAC) cohort [24]. Miettola and colleagues carried out another relevant study in 2016 [25]. Two hundred and eighty-three mother–offspring pairs were included from a cohort initially designed to investigate children with a very low birth weight who were born prematurely (VLBW) and children born at term (this group is referred to as “Term” in the next section). 

#### 3.6.1. Total Body BMD

Hannam et al. [24] showed a tendency towards an inverse association between preeclampsia and total body BMD, but it was insignificant. Adjustment for confounders was made in four different models: model 1(offspring age and gender), model 2 (model 1 + maternal; smoking, socioeconomic status, age, and parity), model 3 (model 2 + maternal BMI, offspring; fat mass, lean mass, and height), and model 4 (model 3 + birth weight and gestational age).

The total body BMD was greater in offspring exposed to gestational hypertension compared to those who were not when only adjusted for offspring age and gender, but the association seemed to be explained by the maternal BMI and offspring adiposity and size, since statistical adjustment was lost after adjustment for these factors [24].

In the Helsinki study [25], the between-group mean differences were estimated with multiple linear regression. This demonstrated that VLBW offspring from mothers with preeclampsia had a significantly higher total body BMD *Z*-score [25]. Adjustment for current offspring height and BMI and socioeconomic position attenuated the difference. The analysis was also adjusted for offspring physical activity, current smoking, maternal smoking, parity, and pre-pregnancy BMI, and, where available (only in 237 pairs), it was also adjusted for current offspring calcium, phosphate, and vitamin D intake, but this had little effect on the results. In Term offspring, there were higher whole-body BMD *Z*-scores among offspring from preeclamptic mothers compared to offspring born from non-preeclamptic. Adjustment for different confounders did not change the results.

In the Helsinki study [25], an analysis of maternal GH was performed, but only five offspring were exposed to GH in the VLBW group, which was not enough for analysis. In the Term offspring group, 21 were exposed to GH, and these offspring had a similar BMD to those not exposed to GH or PE. 

#### 3.6.2. Lumbar Spine BMD

Hannam et al. found no association between preeclampsia and lumbar spine BMD. On the contrary, lumbar spine BMD was greater in the offspring exposed to GH compared to no HDP; this association was seen when adjusted in model 1 and model 2, as defined in Section 2.1. However, these associations became insignificant after adjustment for maternal BMI and offspring adiposity and height in model 3 [24].

The contrary tendency was found in the other included cohort study [25], where VLBW offspring from mothers who had PE had a higher BMD *Z*-score in the lumbar spine, and it remained after adjustments for confounders. There was also a higher spine BMD *Z*-score among the Term offspring exposed to PE, but this association no longer remained in the first three adjusted models (model 1: gender; model 2: model 1 + offspring current height; model 3: model 2 + offspring current BMI). The further adjusted models (model 3 + parental education, offspring physical activity, offspring smoking, and maternal smoking during pregnancy) showed a significant association between maternal PE and lumbar spine BMD in Term offspring as well.

#### 3.6.3. Femoral Neck/Total Hip BMD

The English study [24] used a multivariable linear regression analysis to determine associations between mothers with PE and GH in comparison with the reference category of no HDP and offspring bone health. The total hip BMD was lower in offspring exposed to preeclampsia when the data were adjusted for only age and gender. The association did not change markedly when adjusted with model 2, 3, or 4 (Table 4). The total hip BMD was greater in the offspring exposed to GH, compared to no HDP, adjusted for confounders in model 1 and model 2. This positive association seemed to be explained by maternal BMI and offspring adiposity and height (when adjusted in model 3) (data not shown in the original article) [24]. 

In the Helsinki study, VLBW offspring [25] born to mothers with preeclampsia had significantly higher BMD *Z*-scores of the femoral neck. Term offspring had higher femoral neck BMD *Z*-scores when born from preeclamptic mothers, compared to offspring who were born from non-preeclamptic mothers. Adjustment for different confounders did not change the results. Data are presented in Table 4. 

### 3.7. Maternal Age 

Only one study evaluated the association between maternal age at birth and bone mass. Rüdang et al. included 1009 male offspring from the GOOD cohort [30]. The GOOD cohort was initiated to determine both the environmental and genetic factors involved in the regulation of bone and fat mass. Rüdang and colleagues showed an inverse association between maternal age and several of the bone parameters measured, but only the results for BMD are presented in this review (refer to Table 5).

#### 3.7.1. Total Body aBMD

Rüdang and his collaboratives [30] found maternal age to be inversely correlated to aBMD in the total body (r = −0.070, *p* = 0.03). However, in the fully adjusted regression analysis, the association was no longer sustained (adjusted for offspring factors (present smoking, calcium intake, current level of physical activity, adult height and weight, birth height, total body adipose tissue, and lean mass) and adjusted for maternal factors (length of pregnancy, socioeconomic index, parity, smoking, weight before pregnancy, height, and parental age)). Data are shown in Table 5.

#### 3.7.2. Lumbar Spine aBMD

Maternal age was also inversely correlated to aBMD at the lumbar spine (r = −0.092, *p* < 0.01). In the fully adjusted regression analysis, the inverse relationship was sustained (*n* = 705, β-coefficients = −0.091, *p* < 0.001). Data are shown in Table 5.

In one of the linear regression models, a decrease in the lumbar spine aBMD at 0.00233 g/cm^2^ with every year increase in maternal age was demonstrated. 

The inverse relationship between maternal age and aBMD in the lumbar spine was found in both smoking and non-smoking offspring (adjusted for physical activity, total body lean mass, and adult height) [30].

#### 3.7.3. Femoral Neck BMD

There was no correlation between maternal age and aBMD at the femoral neck [30]. 

#### 3.7.4. Subsample Analysis of Maternal Age

In a subgroup analysis, the mothers were divided into two groups, where the oldest mothers (>36 years, corresponding to the 90th percentile of age) were compared to the younger mothers (<36 years). BMD was adjusted for covariates correlated to aBMD (offspring factors: total body lean mass, total body fat mass, current smoking, calcium intake, current physical activity, adult height, adult weight, birth height, and length of pregnancy). After that, the two groups were compared with an independent sample *t*-test. The offspring of the oldest mothers had a lower BMD in the total body (1.6%), lumbar spine (2.6%), and femoral neck (2.8%) than offspring from the younger mothers [30]. 

### 3.8. Maternal Cigarette Smoking

We identified two studies evaluating the effect of smoking during pregnancy on offspring bone health, results are summarized in Table 6. Jones et al. [26] examined the relationship between maternal cigarette smoking during pregnancy and offspring bone mass at age 16 in Tasmania. The 415 adolescents included was initially part of a cohort investigating sudden infant death syndrome (SIDS). The 415 adolescents represent 29% of the original cohort. 

Another study evaluated the association between exposure to intrauterine smoking and the offspring BMD and BMC at the age of 18 years old and was performed by Martínez-Mesa and collaborators. The adolescents were part of the Pelotas birth cohort in Southern Brazil, comprising 5249 newborns. Of these, 3855 participated in the 18-year follow-up visit and had a whole-body DXA obtained [27], but only 3075 were included in the analysis. 

#### 3.8.1. Total Body BMD

Through a multivariable linear model (adjusted for offspring factors (current height, weight, age, gender), the season of birth, gender, maternal age, the duration of the second stage of labor, and maternal intention to breastfeed), Jones et al. [26] did not find an association between maternal smoking during pregnancy and total body BMD; see Table 6.

In the Brazilian study [27], there was a tendency towards an inverse association between maternal smoking during pregnancy and the offspring BMD at age 18. The authors used a linear regression analysis resulting in a β-coefficient per each additional cigarette smoked during pregnancy. The regression analysis was adjusted by maternal factors (partner smoking, gestational age, height, age, skin color, education, income) plus offspring factors (smoking, physical activity status, alcohol consumption, and calcium intake) (adjusted by total calorie consumption). The study wanted to assess if maternal smoking had an indirect effect on BMD by mediation through birth weight and contemporaneous anthropometric measures (height, weight, and BMI). It seemed so since the week inverse association between maternal smoking and bone health was even more attenuated when accounting for these factors. All the data are presented in Table 6.

#### 3.8.2. Femoral Neck BMD

Only the Tasmanian study had femoral neck BMD as an outcome, and no association between smoking during pregnancy and femoral neck BMD was found [26].

#### 3.8.3. Lumbar Spine

Jones and colleagues did not report an association between smoking during pregnancy and spine BMD in the offspring at age 16 [26].

#### 3.8.4. Fractures

Jones et al. also investigated the association between fractures and BMD. The study did not find an association between smoking in pregnancy and fractures in offspring at age 16 [26].

## 4. Discussion

In this systematic review, there was a paucity of high-quality studies, but there is an indication that certain nutritional factors in pregnancy may be beneficial in terms of achieving a higher peak bone mass among the offspring. Specifically, high concentrations of 25(OH)D during pregnancy; low maternal fat intake; and high intakes of calcium, phosphorus, and magnesium during pregnancy all may increase the BMD in offspring as young adults. Moreover, the evidence suggests that the offspring of younger mothers may have a higher peak bone mass compared to those of older mothers. Whether there is influence of preeclampsia or maternal smoking on bone health among young adults remains inconclusive. 

In context with previous published observational evidence [7,8,32,33,34,35,36,37,38,39], mother–offspring cohorts have suggested that factors such as maternal diet, lifestyle, physical activity, hormones, and vitamin D status during pregnancy were all associated with bone development in the offspring at birth, before puberty, or during adolescence, which may support the findings of this review.

The certainty of the evidence base included was very low, due to serious risk of bias, because the included studies did not adjust for all our prespecified confounders (age, gender, ethnicity, and maternal BMI. 

Moreover, there is a high risk of over-adjustment bias in some of the analyses of the statistical models in the included studies, as they included covariates that were measured postnatally—i.e., breastfeeding, sports participation, and anthropometric measures of the offspring—and did not temporally precede the exposure.As such, these covariates should rather be considered mediators, and thus incorporating these in the model may create a distortion of the results. At this point, future studies are recommended to present a causal directed acyclic graph to document the rationale for the adjustments in the statistical models [40].

Furthermore, the imprecise measurement of self-reported lifestyle variables is well-known—i.e., the over-reporting of healthy lifestyle or under-reporting of an unhealthy lifestyle—and thus the studies on dietary intake and cigarette smoking during pregnancy are subject to risk of bias related to misclassification, and this is likely to distort the estimation of the results.

That being said, none of the included studies were categorized as having a critical risk of bias within any of the domains, but all the studies had serious risk of bias in the overall judgment assessed by the ROBINS-I tool. Besides confounding and the risk of misclassification of exposure, the study quality was most frequently challenged within the domains of missing data, and partly in the selection of reported results. The consequences of these methodological challenges are unclear, and both increased risk of type I or II error in the associations shown are likely. This led us to conclude that the overall quality of evidence is very low, which warrants a cautious interpretation of these results.

None of the seven included studies incorporate data on maternal health—i.e., osteoporosis or other relevant comorbidities—with possible effects on offspring bone health. 

Regarding the pregnancy-related complications such as preeclampsia, we identified two studies examining the influence of exposure to preeclampsia in utero and later BMD in the offspring [24,25], but the two studies showed opposing results. We suspect that the findings in the study from Miettola et al. may be subject to chance finding and a high risk of over-adjustment by including offspring covariates that were measured postnatally, and by adding both offspring height and BMI in the same statistical model. Consequently, more studies are needed on the subject.

We identified two studies examining the association between in utero smoking exposure and offspring bone health [26,27]. Our review focused on BMD in the offspring, and no such significant association was found with maternal cigarette smoking during pregnancy. However, Martínez-Mesa et al. [27] found an interesting overall negative association with BMC in male offspring only (*p* = 0.048, BMC results not shown in the tables). This negative association in males disappeared after accounting for birth weight and concurrent anthropometric data in the offspring. It is important for context that existing studies have shown an association between maternal smoking during pregnancy and low birth weight [41,42], and this study also corroborated this. Maternal smoking, therefore, seems to have a long-term negative effect on bone health that is mediated through the low birth weight. The findings of low BMC in young adults born to smoker mothers could indicate that the offspring have smaller bone dimensions and perhaps an overall delayed bone age, or possibly even a permanently reduced PBM when expressed as BMC rather than a real BMD.

### 4.1. Directions for Future Research

We were surprised at the absence of studies that examined a range of other factors that could potentially influence offspring bone health in a life course perspective. Among the factors that have not been adequately addressed, except for in animal models, are prenatal exposure to, e.g., infections (e.g., cystitis); prolonged pregnancy; hyperemesis gravidarum; maternal use of medication during pregnancy, i.e., corticosteroid [43,44,45,46] or selective serotonin reuptake inhibitors [47]; endocrine disruptive chemicals [48,49,50,51]; and estradiol concentrations [52,53]. Moreover, we did not detect any studies that examined the influence of either insulin resistance during pregnancy or gestational diabetes on bone health among adult offspring. Results from several studies in human infants and animal pups suggest a decreased level of bone mineralization at birth in the offspring of diabetic mothers [38,54], and one study in rats suggested that the adverse effect of gestational diabetes on bone in male rats persisted into young adulthood [55].

This major research gap calls for an in-depth examination of the influence of early life determinants on the timing and magnitude of peak bone mass. The characterization of such determinants will inform recommendations and strategies aimed at improving bone development to peak, with the long-term aim of lowering the prevalence of fractures amongst the elderly.

### 4.2. Strength and Limitations of the Systematic Review

The main strength of this review lies with our systematic approach and quality assessment of the included studies. The protocol was registered in advance in PROSPERO before initiating the work, and to ensure a high quality, the study selection and quality assessment were performed independently by two researchers. We performed a comprehensive literature search; however, for pragmatic reasons, we applied pre-specified limitations to the search, which may have narrowed our search result, leaving out relevant studies. Studies within the subject of preeclampsia and maternal smoking were not identified in our search and were subsequently identified through contact with content experts. We suspect that the main reason for the missed references may be related to how the studies were indexed in the databases applied. 

While in the protocol, the age in offspring was set to 18–30 years of age, this age span was extended due to minimal publications in the area. Additionally, the exact timing for PBM is not quite precise, but in general it is accepted that most of the skeletal mass is acquired by the age of 20 and continues to grow until the third decade of life [56]. In the meantime, recent studies have shown that PBM differentiates in bone measurements in various bone sites and that peak bone density occurs later in boys than girls, especially in the spine [57].

The paucity of the included studies made the conduction of a meta-analysis infeasible. We used vote counting as a synthesis method. Thus, the conclusions made are based on whether there was any evidence of an effect or association, rather than evaluating the average summary estimate.

## 5. Conclusions

In conclusion, this systematic review highlights inconsistent findings on the association between various prenatal environmental exposures and bone health in early adulthood in offspring. When considering the synthesis of the results in context with the previous published observational evidence on bone health among newborns, children, and adolescents, there may be an indication that maternal diet during pregnancy indeed influences the peak bone mass. However, this conclusion is speculative because of the scarcity of the evidence base, and the study quality was challenged by confounding, the risk of the misclassification of self-reported exposures, missing data, and partly in the selection of the reported results. This led us to conclude that the overall quality of evidence is very low; thus, the assumptions drawn from this sparsely investigated research field hold a high degree of uncertainty. More high-quality studies with sufficient statistical power are needed on the subject, and the inclusion of more studies is likely to change future conclusions.

## Figures and Tables

**Figure 1 nutrients-12-02866-f001:**
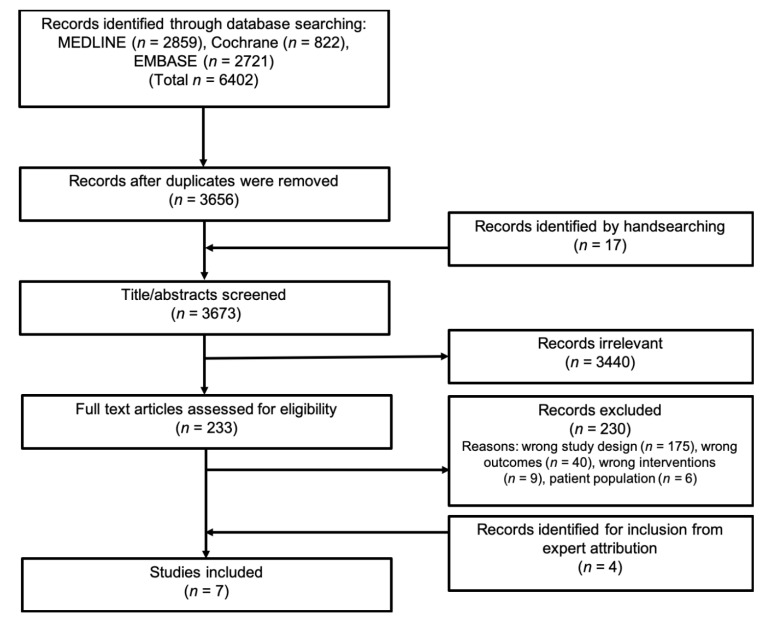
PRISMA flow diagram presenting data from the search and study selection process.

**Table 1 nutrients-12-02866-t001:** Characteristics of the included studies.

1st Author, Year, Country	Study Design	Sample Size (*n*)	Age, Offspring (Mean ± SD)	Maternal Age, at Birth (years)	Exposure, Cointervention	Outcome
**Hannam** [24], **2015,** **England**	Prospective cohort	3088	^1^ PE: 17.9 ± 0.5 GH: 17.8 ± 0.4 No HDP: 17.8 ± 0.4	15–19 *y*: *n* = 36 20–24 *y*: *n* = 367 25–29 *y*: *n* = 1165 30–34 *y*: *n* = 1101 >40 *y*: *n* = 51	Preeclampsia and gestational hypertension. No co-intervention.	BMD
**Jones** [26], **2013,** **Tasmania**	Prospective cohort	415	^2^ Never breastfeed: 16.3 ± 0.5 Ever breastfeed: 16.3 ± 0.4	(Mean ±SD) Offspring never breastfeed: 24.1 ± 5.0 Offspring ever breastfed: 26.2 ± 4.6	^3^ Maternal smoking during pregnancy. No cointervention	BMD, fractures
**Martínez-Mesa** [27], **2014,** **Southern Brazil**	Prospective cohort	3075	18 years	(Mean (s.e)) Offspring male: 26.5 (0.17) Offspring female: 26.5 (0.15)	^4^ Maternal smoking during pregnancy. No co-intervention	BMD, BMC
**Miettola** [25], **2013,** **Finland**	Prospective cohort	5283	^5^ VLBW: ^6^ PE 22.5 ± 2.0 No PE 22.6 ± 2.2 ^5^ Term: PE 21.5 ± 1.7 No PE 22.7 ± 2.2	-	Preeclampsia no co-intervention	BMD, BMAD, BMC
**Rudäng** [30], **2012,** **Sweden**	Prospective cohort	1068	18.9 ± 0.6	(Mean ±SD) 29.5 ± 4.8	Maternal age. No co-intervention	aBMD, BMC, BA, CSA, endosteal and periosteal circumference, trabecular and cortical vBMD
**Yin** [28], **2010,** **Tasmania**	Prospective cohort	216	16.2 ± 0.4	(Mean ±SD) 26.7 ± 4.7	^7^ Maternal dietary intake during the third trimester. No co-intervention	BMD, BMC
**Zhu** [29], **2014,** **Australia**	Prospective cohort	341	Male: 20.2 ± 0.5 Female: 20.1 ± 0.5	(Mean ±SD) 29.0 ± 5.6	Serum 25-hydroxyvitamin D during pregnancy. No co-intervention	BMD, BMC, BA

*n*, number; *y*, years old; GH, gestational hypertension; PE, preeclampsia; No HDP, no hypertensive disorders of pregnancy; BMD, bone mineral density; BMC, bone mineral content; VLBW, very low birth weight; Term, offspring born at term; BMAD, bone mineral apparent density; aBMD, areal bone mineral density; BA, bone area; CSA, cortical cross-sectional area; vBMD, volumetric bone mineral density; ^1^ PE, the definition is systolic blood pressure >139 mmHg or diastolic blood pressure >89 mmHg, at least once after 20 weeks of gestational age and proteinuria on dipstick ≥1 (30 mg/dL); GH was not included in the PE group, preexisting hypertension was not included in the PE or GH group. ^2^ Study participants were split into two groups, ever breastfed and never breastfed, according to maternal recall about breastfeeding, obtained when the child was eight years old. ^3^ Maternal smoking in any trimester of pregnancy was measured by a postnatal questionnaire while the mother and baby were in the hospital. ^4^ Number of cigarettes smoked per day was reported by mothers in a questionnaire and analyzed as a continuous variable. ^5^ A total of 144 participants called VLBW (preterm with very low birth weight, <1500 g) and 139 controls called Term (matched according to gender, from all consecutive births in the same hospital and who were not small for gestational age). ^6^ PE: blood pressure >140/90 mmHg after mid-pregnancy, and proteinuria ≥0.3 g protein excretion in a 24 h urine sample or a positive dipstick, no history of hypertension medication before pregnancy. ^7^ i.e., protein, fat, carbohydrate, calcium, magnesium, phosphorus, fish, fruit, milk, meat, and vegetables. Please refer to Appendix A for additional study details.

**Table 2 nutrients-12-02866-t002:** Summary table of the risk of bias in the included studies evaluated using “Risk Of Bias In Non-randomized Studies of Interventions” (ROBINS-I). The overall judgment of risk of bias equals the most severe level of bias found in any domain.

1st Author	Bias Due to Confound-ding	Bias Due to Selection of Participants into the Study	Bias in Classification of Interventions	Bias Due to Departures from Intended Interventions	Bias Due to Missing Data	Bias in Measurement of Outcomes	Bias in Selection of Reported Results	Overall Judgment
**Hannam** [24]	Moderate	Low	Moderate	Low	Serious	Low	Moderate	Serious
**Jones** [26]	Serious	Serious	Moderate	Low	Serious	Moderate	Moderate	Serious
**Martínez-Mesa** [27]	Serious	Low	Low	Low	Serious	Low	Moderate	Serious
**Miettola** [25]	Serious	Low	Low	Low	Serious	Low	Serious	Serious
**Rudäng** [30]	Serious	Low	Low	Low	Serious	Low	Moderate	Serious
**Yin** [28]	Serious	Serious	Moderate	No information	Serious	Moderate	Moderate	Serious
**Zhu** [29]	Moderate	Low	Moderate	No information	Serious	Moderate	Moderate	Serious

**Table 3 nutrients-12-02866-t003:** Summary table of results from the articles evaluating the association between the offspring bone mineral density, maternal nutrient intake, and vitamin D status during pregnancy.

	Outcome	Total Body BMD	Lumbar Spine/Spine BMD	Femoral/Total Hip BMD
1st Author				
**Yin** [28]		**Food groups**		
		BMD not affected by maternal meat density ^1^	BMD not affected by maternal meat density ^1^	BMD not affected by maternal meat density ^1^
		BMD not affected by maternal fish density ^1^	BMD not affected by maternal fish density ^1^	BMD not affected by maternal fish density ^1^
		BMD not affected by maternal milk density ^1^	BMD increases with increasing maternal milk density ^1^ (mL/kJ) ^2^β: +0.41 (r^2^ 0.213) (*p* < 0.05)	BMD not affected by maternal milk density ^1^
		BMD not affected by maternal vegetable density ^1^	BMD not affected by maternal vegetable density ^1^	BMD not affected by maternal vegetable density ^1^
		BMD not affected by maternal fruit density ^1^	BMD not affected by maternal fruit density ^1^	BMD not affected by maternal fruit density ^1^
		**Macronutrients**		
		BMD not affected by maternal protein density ^1^	BMD not affected by maternal protein density ^1^	BMD not affected by maternal protein density ^1^
		BMD not affected by maternal fat density ^1^	BMD declines with increased maternal fat density ^1^ (g/kJ) ^2^β: −10.3 (r^2^ 0.217) (*p* < 0.05)	BMD declines with increased maternal fat density ^1^ (g/kJ) ^2^β: −11.3 (r^2^ 0.366) (*p* < 0.05)
		BMD not affected by maternal carbohydrate density ^1^	BMD not affected by maternal carbohydrate density ^1^	BMD not affected by maternal carbohydrate density ^1^
		**Micronutrients**		
		BMD not affected by maternal calcium density ^1^	BMD increases with increasing maternal calcium density ^1^ (mg/kJ) ^2^ β: +0.36 (r^2^ 0.216) (*p* < 0.05)	BMD not affected by maternal calcium density ^1^
		BMD not affected by maternal magnesium density ^1^	BMD increases with increasing maternal magnesium density (mg/kJ) ^2^ β: +2.9 (r^2^ 0.217) (*p* < 0.05)	Some of the analysis indicated increasing BMD with increasing maternal magnesium intake
		BMD not affected by maternal phosphorus density ^1^	BMD not affected by maternal phosphorus density ^1^	BMD not affected by maternal phosphorus density ^1^
**Zhu** [29]		BMD increases with higher maternal serum 25OHD concentration ^3^ Mean, 95% CI: 4.6 [0.1;9.1] mg/cm^2^	-	-
		BMD declines with maternal vitamin D deficiency ^4^ Mean ± SD: 1053 ± 7 versus 1071 ± 5 mg/cm^2^ (*p* = 0.043)	-	-

BMD, bone mineral density; 25OHD, 25-hydroxyvitamin D; CI, confidence interval. Study results from the fully adjusted analysis are presented, and only significant data is shown. ^1 ^Every dietary variable was converted to density measures by dividing estimated daily nutrient intake by estimated total daily energy intake. ^2^ β-coefficients for the association between maternal nutrient intake during pregnancy and BMD (g/cm^2^) assessed with a linear regression analysis. Data adjusted for potential confounders—i.e., offspring factors (gender, weight at age 16, sunlight exposure in the winter, sports participation, current calcium intake, Tanner stage at age 16, ever breastfed) plus maternal factors (smoking during pregnancy, age at the time of childbirth, and the different dietary instruments used). ^3^ Expected differences (mean, 95% CI) in offspring BMD with each additional 10 nmol/L of serum 25OHD during pregnancy (found in a linear regression analysis), adjusted for the season of maternal blood sample collection and offspring gender, age at DXA, maternal education, parity, ethnic origin, maternal height and weight before pregnancy, offspring birth weight, height, lean mass, and fat mass at age 20 years. ^4 ^Expected means (by analysis of covariance) in offspring BMD in mothers with serum 25OHD <50 nmol/L vs. ≥50 nmol/L at 18 weeks of pregnancy. It is adjusted for the season of maternal blood sample collection and offspring gender, age at DXA, maternal education, parity, ethnic origin, maternal height and weight before pregnancy, offspring birth weight, height, lean mass, and fat mass at age 20 years.

**Table 4 nutrients-12-02866-t004:** Association between maternal preeclampsia and gestational hypertension with the offspring bone mineral density (BMD) in the total body, spine, and hip.

	Outcome	Total Body BMD	Lumbar Spine BMD	Femoral/Total Hip BMD
1st author				
**Hannam** [24]		No association between BMD and PE	No association between BMD and PE	BMD was inversely associated with PE ^1^ Mean difference (95%CI): −0.30 (−0.50 to −0.10), *p* = 0.004
		No association between BMD and GH in the fully adjusted data	No association between BMD and GH	No association between BMD and GH in the fully adjusted data
**Miettola** [25]		Preeclampsia had a direct association with BMD in VLBW offspring. ^2^ Mean difference(95%CI): 0.46(0.15–0.76) *p* = 0.003	Preeclampsia had a direct association with BMD in VLBW offspring ^2^ Mean difference(95%CI): 0.42(0.08–0.76), *p* = 0.016	Preeclampsia had a direct association with BMD in VLBW offspring ^2^ Mean difference(95%CI): 0.37(0.06–0.68), *p* = 0.020
		Preeclampsia had a direct association with BMD in Term offspring ^2^ Mean difference (95%CI): 0.87(0.28–1.46), *p* = 0.004	There was a direct association between BMD and PE in Term offspring in the fully adjusted model, but not in the first 3 models ^3^. ^2^ Mean difference(95%CI): 0.70(0.13–1.27), *p* = 0.017	Preeclampsia had a direct association with BMD in Term offspring ^2^ Mean difference (95%CI): 0.70 (0.17–1.23), *p* = 0.010
		No association between GH in Term offspring and BMD. No analysis was performed for VLBW offspring.		

BMD, bone mineral density; PE, preeclampsia; GH, gestational hypertension; No HDP, No hypertensive disorder of pregnancy; VLBW, very low birth weight and born prematurely; Term, offspring born at term; No PE, no preeclampsia; CI, confidence interval. The table presents fully adjusted results, and only significant data is shown. ^1^ Multivariable linear regression analysis adjusted for offspring age at scan, gender, fat mass, lean mass, height, birth weight, and gestational age and maternal smoking, socioeconomic status, maternal age, parity, and BMI. ^2 ^The group mean differences were estimated with multiple linear regression. The data shown are adjusted for offspring gender, current offspring height, current offspring body mass index, parental education, offspring physical activity, offspring smoking, and maternal smoking during pregnancy. There were missing data for 12 cases in the VLBW group and, therefore, they were not included in the fully adjusted model. There were missing data for two patients in the Term group, and they were not included in the fully adjusted model shown here. ^3^ Model 1: gender; model 2: model 1+ offspring current height; model 3: model 2 + offspring current BMI; model 4: model 3 + parental education; model 5: model 4 + offspring physical activity, offspring smoking, and maternal smoking during pregnancy.

**Table 5 nutrients-12-02866-t005:** Association between the maternal age at birth and the offspring bone mineral density in adolescents.

	Outcome	Total Body aBMD (g/cm^2^)	Lumbar Spine aBMD (g/cm^2^)	Femoral Neck aBMD (g/cm^2^)
1st author				
**Rudäng** [30]		aBMD declines with increasing maternal age in a bivariate correlation but become non-significant in the stepwise linear regression model β: −0.031 (NS) r value: −0.070 (*p* < 0.03)	aBMD declines with increasing maternal age: β: −0.091(*p* < 0.01) r value: −0.092 (*p* < 0.01)	Femoral aBMD was not affected by maternal age.

aBMD, areal bone mineral density; NS, not significant. Results from the fully adjusted analysis are presented, and only significant data is shown. Standardized β-coefficients were assessed using a stepwise linear regression model. Bivariate correlation with maternal age was evaluated using Pearson’s correlation, the r-value is presented. Data are adjusted for offspring calcium intake, current level of physical activity, adult height and weight, birth height, total body adipose tissue and lean mass, length of pregnancy, present smoking in the offspring, socioeconomic index (SEI), maternal parity, maternal smoking, paternal age, maternal weight before pregnancy, and maternal height.

**Table 6 nutrients-12-02866-t006:** Association between maternal cigarette smoking during pregnancy and offspring bone mineral density in adolescence.

	Outcome	BMD
1st author		
**^1^****Jones** [26]		Total body BMD, lumbar spine BMD, and hip BMD Not affected by maternal smoking during pregnancy.
**^2^****Martínez-Mesa** [27]		Total body BMD not affected by maternal smoking in an overall association
		Total body BMD not affected by maternal smoking after accounting for mediation by birth weight and concurrent height
		Total body BMD not affected by maternal smoking after accounting for mediation by birth weight and concurrent weight
		Total body BMD not affected by maternal smoking after accounting for mediation by birth weight and concurrent BMI

BMD, bone mineral density (g/cm^2^); BMI, body mass index (kg/m^2^). Only results from the fully adjusted analysis are presented. None of the results were significant. ^1^ Jones et al. assessed the multivariate association between smoking in utero (yes vs. no) and BMD (g/cm^2^) as β-coefficients (95% CI). Data were adjusted for current offspring height, weight, age, gender, season of birth, gender, maternal age, duration of the second stage of labor, and maternal intention to breastfeed. ^2^ Martínez-Mesa et al.: The association between offspring BMD and maternal smoking in pregnancy was assessed with the linear regression coefficient, with each additional cigarettes smoked during pregnancy indicated as β-coefficients (95% CI); results are segregated in males/females. All the presented data were adjusted by partner smoking, gestational age, maternal height, maternal age, maternal skin color, maternal education, income, and offspring smoking, physical activity status, alcohol consumption, and calcium intake (adjusted by total calorie consumption). Further, the investigators evaluated if maternal smoking had an indirect effect on BMD by mediation through birth weight and contemporaneous anthropometric measures.

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
