# Peer review of "Nutrients, Diet, and Other Factors in Prenatal Life and Bone Health in Young Adults: A Systematic Review of Longitudinal Studies"

_nutrients, 2020, doi:10.3390/nu12092866_

Round 1

Reviewer 1 Report

The article presents a good bibliographic review and the appropriate review process has been carried out and the appropriate conclusions have been obtained.

The work carried out has more than met the essential requirements for a systematic review: protocol registration, search in all relevant bibliographic databases, PRISMA process, etc. It is very interesting, due to its repercussions for later health, to know the gestational factors that intervene in the life of the later adult; in this sense, studies are necessary to elucidate these factors with great scientific rigor. This article highlights what is already known and above all what cannot be concluded at this time and what it is necessary to deepen. This point is perhaps the one of greatest interest in the article as it highlights the need for further research on maternal factors related to bone mineralization in adults.

Author Response

On behalf of all authors, we would like to thank you for your time reviewing our systematic review.

Reviewer 2 Report

The article presented is interesting and describes in a clear and concise way the objective of the study. However, a minor revision of the document would be necessary.

Abstract:
Line 22. Define ‘BMD’

Introduction:
Line 59. Define ‘BMD’ and remove italics in the text 'in early adulthood'

Material and Methods.
Line 99. Define ‘BMC, BA, PBM’

Table 1. Drop column of conflict of interest and cite this item in the text, because the seven papers no have conflict of interest. I recommend to modify the format of table with reduction of size font to view better the text.

Table 2. Modify the format. The table is interesting but it is badly displayed. Perhaps it would be advisable to modify its format by reducing the title of the columns or to make a colour scale that allows a better visualization since there are few studies.

Table 3. Add number of study’s reference in the table.

Lines 226-227. To do reference in the text of Table 3.

Table 4. Modify format. The blue color is out of table in the bottom. Add number of study’s reference in the table.

Line 315. Add number of reference ‘In the Helsinki study, …’

Table 5. Add number of study’s reference in the table.

Table 6. Add number of study’s reference in the table.

Author Response

Detailed changes in Manuscript: nutrients-939207

On behalf of all authors, we would like to thank you for your time reviewing our systematic review. We fully agree with the suggestions made by the reviewers, so we are happy to inform you that we were able to accommodate all comments. Below is a detailed description of revisions made in the manuscript:

Abstract:
Line 22: ‘BMD’ changed to “bone mineral density”

Introduction:
Line 59. ‘BMD’ defined as “bone mineral density” and italics removed in the text 'in early adulthood'

Material and Methods:
Line 100-101: BMC defined as “Bone mineral content”, BA defined as “Bone area”, PBM defines as “Peak bone mass”,

Table 1: The column of “conflict of interest” is deleted. The column “Calibrated (+) / No information (NI)” deleted and added to table S4. Size reduction of the table and colours changed.

Material and Methods.

Line 158: In compensation of the deleted column of “conflict of interest” in table 1 sentence added: “None of the seven included studies had conflict of interest to declare.”

Table 2: Format modified; Change of colours and reduction of size.

Table 3: Number of study’s reference added. Format modified; Change of colours and reduction of size.

Results

Lines 226-227. To do reference in the text of Table 3.

Table 4: Format modified; Change of colours and reduction of size. and number of study’s reference added

Results:

Line 315. Number of reference added.

Table 5: Number of study’s reference added and format modified; Change of colours and reduction of size.

Table 6: Number of study’s reference added and format modified; Change of colours and reduction of size.

Table S4: The column “Calibrated (+) / No information (NI)” added and format modified; Change of colours and reduction of size plus bold text type in 1st Column.
